# Implementing psychological first aid ontology

Hyun-Ok Jung[1], Seung-Woo Han[2]*

1 College of Nursing, Kyungpook National University, Daegu, Republic of Korea, 2 Department of Emergency Medical Technology, Kyungil University, Gyeongsan, Republic of Korea

* swhan@kiu.ac.kr

## Abstract

### Background

This study develops an ontology of Psychological First Aid (PFA) by extracting relevant knowledge from a review of PFA literature.

### Materials and methods

This study was conducted using the PFA ontology development 101 method. This review processes previously-developed PFA studies by consulting Google Scholar, CINHL, PUBMED, and MEDLINE. Protege 5.0 program was used to integrate with ontology development. The developed PFA ontology consisted of eight super classes: Action agenda, Assessment, Concrete method, Disaster type, Disaster disposition, Purpose, Qualification and Skill, Reaction. In total, 166 terms were collected.

### Results

The eight super classes were divided into 72 classes and 64 subclasses. The composition yielded in a total of 166 axioms (85 logical axioms; 81 declaration axioms).

### Conclusions

This study provides basic data to guide development and composition of PFA arbitration programs.

**Data Availability Statement:** All relevant data are within the paper and its Supporting Information files.

**Funding:** The authors received no specific funding for this work.

## Introduction

Victims of natural disasters are likely to suffer from psychological shock, pain and extreme stress reactions. If these psychological traumas are not resolved, they may lead to depression, anxiety, and even post-traumatic stress disorder [1]. Psychological First Aid (PFA) is an act of providing immediate psychological support by mental health counselors or mental health professionals to people who need psychological help after a disaster. PFA is distinct from medical procedures such as artificial respiration at the scene of an accident [2]. PFA aims to improve the physical and psychological short-term function by helping victims to regain psychological stability after a disaster, and focuses on reinforcing the assistance of disaster victims by

**Competing interests:** The Authors have had that no competing interests exist.

connecting support systems for continuous help [3]. Therefore, education in PFA should be expanded, manuals and an integrated system established. This process requires understanding and connection of information structure between human and software by implementing an ontology for PFA. An ontology is a representation of the relations among a set of concepts and categories. It can be used to clarify concepts by defining terms in a consistent form that can be used to guide processing of information [4].

Recently, ontology research in the field of health and medical care is mainly used to define and connect diagnosis and medical terms. However, it has been not easy to systematize one information on psychological first aid in emergency rescue, psychology, health science, nursing, and medicine, and to provide PFA to actual subjects through customized services based on a consistent method. Furthermore, methods to provide PFA to actual subjects by using customized services and a consistent method have not been developed. Therefore, the aim of this study was to clarify standards and protocols that can provide PFA to victims in the event of a disaster by implementing PFA ontology.

The purpose of this study is to implement an ontology as an analysis framework for collecting and classifying documents related to PFA, and to us the ontology to develop practical guidelines for PFA.

## Materials and methods

In this work, we use Class, Object, Property, and Data property to collect and classify PFA terms used in the domain, and to associate meaningfully defined classes. The PFA ontology of this study was implemented according to the Ontology Development 101 [5] procedure, which was described to help beginners to understand the development process of health and medical ontology. The process of implementing PFA ontology by using the Ontology Development 101 method is as follows [5].

First, we identified the area to be included in the ontology and the application range. The purpose of this study was to determine (1) the population to which the ontology will be applied, (2) the purpose of the ontology development, and (3) the area in which the PFA will be expressed in this study. One of the ways to answer questions that identify these areas and ranges is to write a competency question [6]. Then resources that can be used in each literature are collected to implement the desired PFA ontology. The terms that will constitute the ontology are collected. This process can clarify the concept in the process of implementing ontology and identify the definition of attributes necessary to establish the relationship between concepts. Then we defined hierarchical relationships among classes.

In this step, terms with the same meaning and concept are classified considering the attributes and nature of terms, then the hierarchical relationship between them is implemented.

In this study, after defining classes, specific concepts were arranged using a top-down technique to establish a classification system between classes, while at the same time establishing comprehensive concepts. Then we defined class attributes. In this step, after defining he hierarchical relationship between classes are defined, we define the attribute of each class, and all lower classes inherit the attributes of the upper classes, and pass them to the still-lower classes. Then we defined facets of attributes. Types of attributes to be included were String, Number, Boolean, Enumerated, and Instance.

A string is a list of letters, such as a name; a Number is a numeral, Boolean is a yes/no format; Enumerated is a specific value of a set (e.g., the degree of symptoms is indicated as 'mild', 'moderate', or 'severe'); Instance an attribute value of a specific class, which means that it becomes an instance of a specific class [7].

## Results

The first step in building an ontology is to determine its domain and scope; i.e., the purpose of using the ontology, the target demographic, and the management group. These questions are useful for testing later ontologies. The following questions are presented in the PFA ontology. Then we searched for existing searchable and reusable ontologies. We utilize Google Scholar, CINHL, PUBMED, and MEDLINE to search previously-developed PFA. The main existing ontologies were used for disaster management and emergency response systems. We used bio-portal, OSL, and ontobe to utilize the databases. Then we collected and enumerated the terms. The search method applied to the database in the literature was "Psychological" AND "First Aid" AND "Ontology". In order to extract the latest literature, we have limited it to papers published over the last 10 years from January 1, 2012 to January 31, 2021. The results of the study were selected for the final seven papers except for non-original article (n = 44), non-relevant outcome (n = 8), non-relevant study (n = 3), non-English paper (n = 1), and multiple publication (n = 1). Researchers applied the search formula to search and review the text of the literature one by one through several meetings and telephone exchanges. This repeated process eliminated literature that did not conform to the research topic. The researchers conducted cross-analysis once again through a meeting to confirm the final seven papers selected, and if there is no consensus between researchers, the final 100% agreement was reached until the agreement was reached through the final agreement (Fig 1).

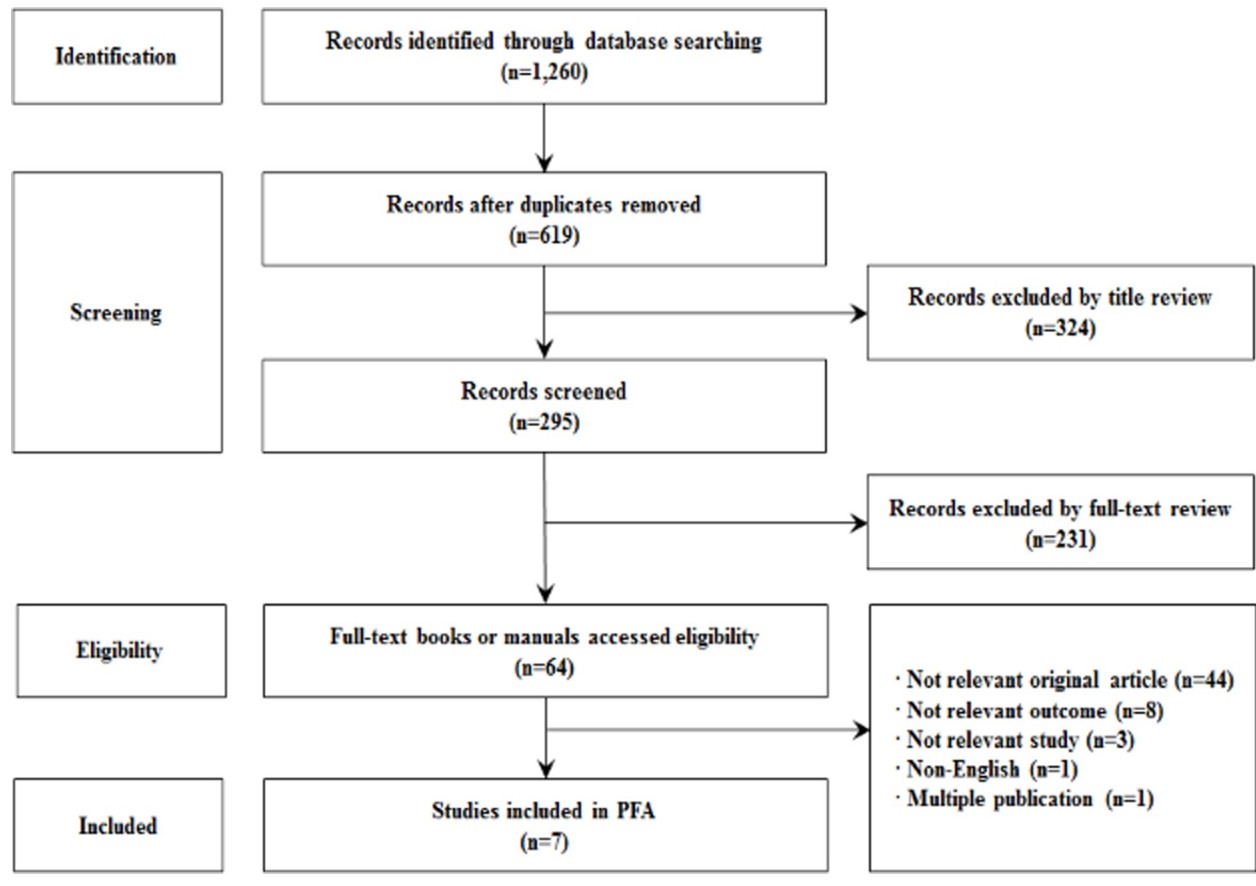

**Fig 1. Flow chart of the sample selection process.**

By scanning the existing literature and papers related to PFA, we focused on establishing a comprehensive list regardless of the overlap between concepts, the attributes of terms, or the relevant class. Next, we defined a hierarchy of classes. We extracted eight superclass concepts: Action agenda, Assessment, Concrete method, Disaster type, Disaster disposition, PFA purpose, Qualification and Skill, and Reaction, then used the top-down developing method to specify the low concepts sequentially. As a result, the PFA ontology was composed of 72 classes, which each consisted of two or three levels (Fig 2).

Following individual cases, the final psychological first aid ontology implementation of this study is as follows (Fig 3).

Finally, after the hierarchical structures between classes had been established, we defined the attributes for each class and described its internal structure.

Any relationship can be established between ontology concepts, but this process can be difficult and complicated to infer if too many relationships are derived. In this study, an independent relationship was maintained in each superclass, and the attribute relationship between the 'Concrete method' and the 'Assessment' was set as the attribute relationship between 'hasmethodof' and 'ismethodof', and the relationship between the action agenda and the core activity was set as 'has a'. We identified 64 subclasses and 72 Classes by consulting the collected papers and manuals. The composition resulted in a total of 166 axioms (85 logical axioms; 81 declaration axioms) [2,8–13] (Table 1).

We set the relationship between 'Action agenda', 'Qualification and Skill' and 'Concrete method' was as 'guidedby'. In this study, the severity was divided into five categories by considering the Impact of Event Scale–Revised (IES-R), which is commonly used to rate event impact. IES-R was classified as subclinical, mild, moderate, or severe (Table 2).

In the end, we set five categories: Class, Property, Value type, Value set, and Cardinality. In particular, the severity of the existing PFA was not included, so the emphasis was placed on setting the PFA severity attribute. At this time, the PFA severity attribute was classified as Severity, and the Value type was set up with a method of selecting a specific one from the set values (Enumerated). The Value set ranged from "No symptoms" to "Extremely high level of symptoms". Cardinality represents means the number of values that an attribute can have [7], and in this study, it was set as 'one' (Table 3).

## Discussion

We extracted eight superclass concepts: Action agenda, Assessment, Concrete method, Disaster type, Disaster disposition, PFA purpose, Qualification and Skill, and Reaction, then used the top-down developing method to specify the low concepts sequentially.

PFA includes measures to stabilize survivors, to provide practical help, and to provide immediate and firm intervention to assist them to get out of the psychological crisis [2].

In this study, the ontology of PFA was set up to express the comprehensive meaning and function of PFA by using three super classes (Nature of disaster, Type of disaster, and Purpose of PFA).

This classification is consistent with the results of a previous study, [2] which indicated that the first step of PFA is to recognize various situations that accompany a disaster, and to consider the purpose of the intervention. Therefore, this attempt to conceptualize the nature and meaning of the PFA is meaningful.

The fourth superclass was divided into Acute response and Post-disaster response. The Acute response was divided into five types: physical, cognitive, emotional, behavioral, and spiritual. The post-disaster responses were divided into four types: shock phase, reaction phase, recovery phase, and reintegration phase. In previous studies, stress associated with trauma was

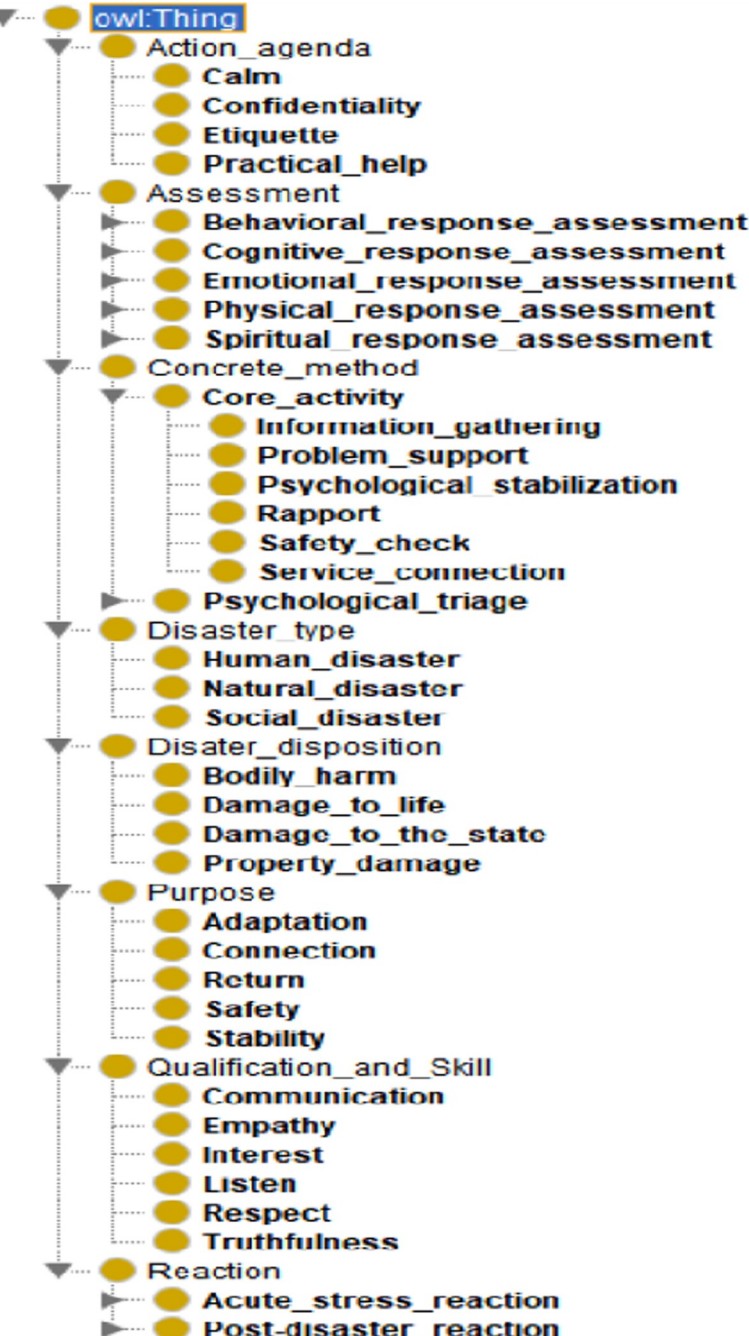

**Fig 2. Class concept hierarchy.**

conceptualized as a dynamic process in which the central nervous system and peripheral nervous system work: they show changes in integrated cognition, emotion, and physiological responses [14] and also show a correlation with faith and religion [15].

In contrast, during the post-disaster response, many behavioral struggles and escape reactions occur during the shock stage, and cognitive function decreases and concentration are disturbed during the reaction stage.

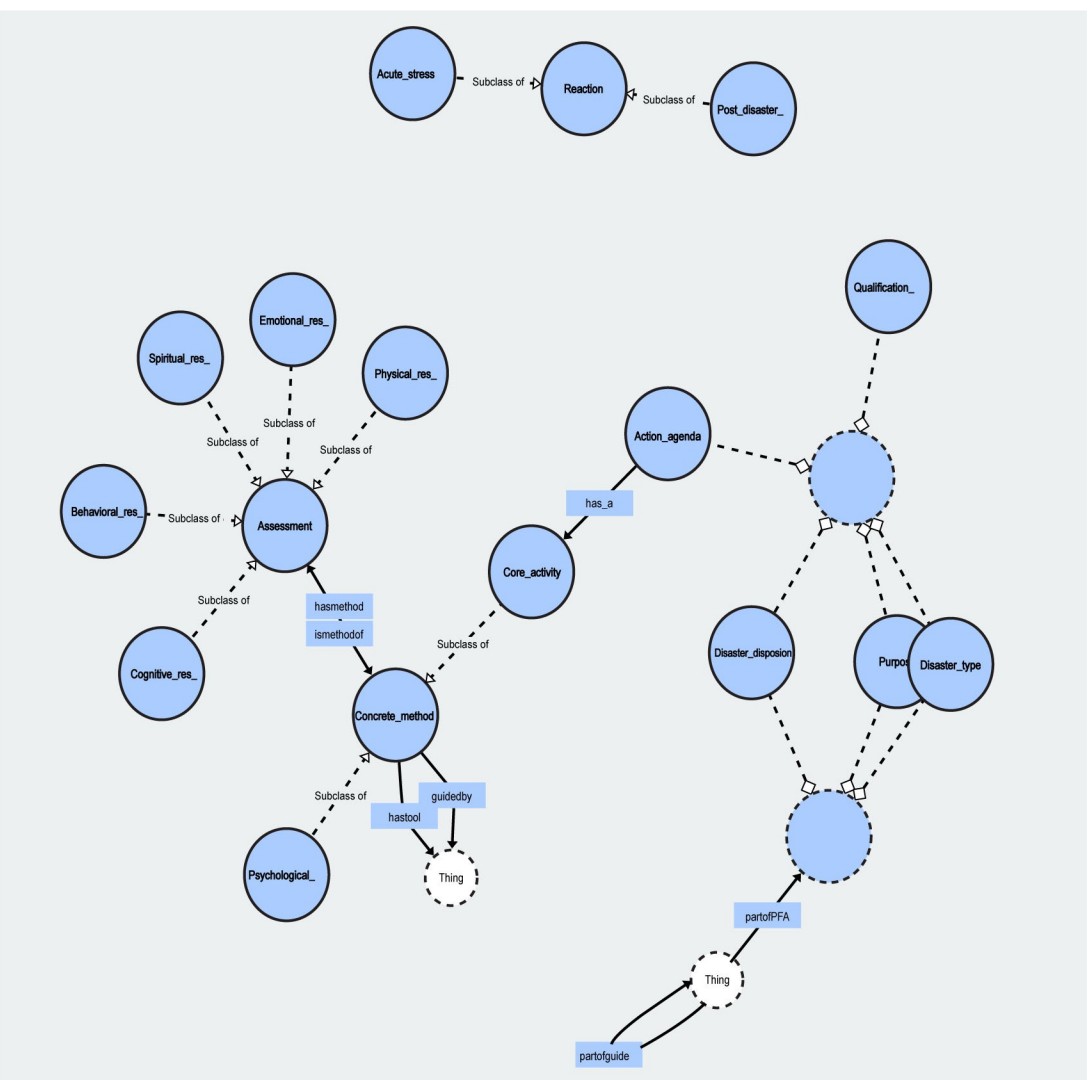

**Fig 3. Psychological first aid ontology implementation.**

**Table 1. Ontology glossary.**

| No | Title | Issued year | Publisher | Author |
|----|-------|-------------|-----------|--------|
| 1 | Disaster and Menntal Health [8] | 2015 | Hakjisa | Kim et al |
| 2 | Handbook for Disaster Nursing and Emergency Prepareness [9] | 2016 | Jenson Books Inc | Veenema, T. G |
| 3 | Psychological first aid for disasters and trauma [10] | 2014 | Hakjisa | Kwon, Ahn, Choi, Joo |
| 4 | Disaster Psychological Support Manual [11] | 2015 | Daegu Disaster Psychological Support Center | Kim |
| 5 | Psychological first aid: Guide for field workers [12] | 2011 | WHO Team Mental Health and Substance Use | WHO, War Trauma Foundation and World Vision International |
| 6 | Facilitation manual: psychological first aid during Ebola virus disease outbreaks [13] | 2014 | World Health Organization | World Health Organization |
| 7 | Psychological first aid. [2] | 2007 | Journal of Mental Health Counseling | Ruzek, J.I. et al |

**Table 2. Relationship design of psychological first aid ontology.**

| Relationship name | Domain | Range | Inverse Relationship |
|---|---|---|---|
| hasmethod | Concrete method | Assessment | ismethodof |
| hasmethod | Assessment | Concrete method | ismethodof |
| has a | Action agenda | Core activity | |
| guidedby | Concrete method | Action agenda Qualification and Skill | |

By this process, during the recovery stage, victims try to improve their adaptive coping ability, and during the reintegration stage, they go through a process of integrating with their own life [11,16].

Therefore, in this study, the response of PFA was classified into acute response and post-disaster response class, and this classification can help to integrate and classify the PFA response. Furthermore, we classified Qualification and Skill, and Action agenda as super classes. We divided Qualification and Skill into empathy, listen, respect, truthfulness, interest, and communication. We divided Action agenda into calm, etiquette, confidence, and practical help. These findings suggest that experts in general qualities and skills [11] should provide victims with a variety of psychological measures, including calm and listening, of which practical help contributes to solving current problems, preventing confusion and facilitating continuous recovery. Therefore, the classification set in this study will help to guide application of PFA to victims.

Finally, we divided the Concrete method superclass into two classes: Psychological triage, and Core activity. We then divided Psychological triage into subclinical, mild, moderate, and severe states. We classified Core activities into six classes: rapport formation, safety, stabilization, information provision, problem solving, and service connection. Generally, Psychological triage PFA has not been included in previous studies; instead, a classification by risk intensity centered on events was common [10], but no classification expressed in psychological reactions and changes.

However, this study is meaningful in that it evaluated and classified accurately and specifically in consideration of the ambiguity of the existing event-centered risk intensity.

We divided the core activities for PFA into (1) Building a sense of trust and stabilizing with the subject, and (2) connecting with various resources. These results are consistent with the results of previous studies in that methods such as building trust and stabilizing the subjects are emphasized as core activities in PFA. Therefore, further studies will need to identify various core activities other than the methods presented in this study.

Finally, in the superclass evaluation part, this study derived five higher concepts (Cognitive evaluation, Emotional evaluation, Physical evaluation, Behavioral evaluation, and Spiritual evaluation). A previous study [10] suggested that to ensure the focus of intervention,

**Table 3. Attribute definition.**

| Class | Property | Value type | Value set | Cardinity |
|---|---|---|---|---|
| Psychological triage | | | | |
| | Psychological severity | | | |
| | | Enumerated | No symptoms | Single |
| | | Enumerated | Few symptoms | Single |
| | | Enumerated | Moderate symptoms | Single |
| | | Enumerated | A High level of symptoms | Single |
| | | Enumerated | An Extremely high level of symptoms | Single |

evaluation is essential, and that the best intervention method, and the safety of survivors by using minimal information, and these areas should consider the physical, psychological, and spiritual dimensions.

This study identified many disaster-related ontologies such as disaster resource ontology and disaster management ontology, but few papers have implemented ontology by characterizing PFA. Therefore, it is meaningful in that it provided an opportunity to characterize PFA in the event of a disaster and to apply it to actual cases.

However, characteristics of disasters can differ among countries (e.g., US: Hurricane, Japan: Earthquake), so application of this ontology to disaster cases in each country may require appropriate modifications. Therefore, future studies should develop ontologies for various types of PFA.

## Conclusions

In this study, PFA ontology was implemented by using Section 101 of Health and Medical Ontology Development (Noy & McGuinness, 2001). Eight super classes were derived: Action agenda, Assessment, Concrete method, Disaster type, Disaster disposition, Purpose, Qualification and Skill, Reaction, then 64 Subclasses and 72 Classes were obtained by reference to the collected papers and manuals. The composition resulted in a total of 166 axioms 85 (Logical axioms; 81 declaration axioms).

## Acknowledgments

The author contributed to implementing PFA based on many prior studies.

## Author Contributions

**Conceptualization:** Hyun-Ok Jung, Seung-Woo Han.

**Data curation:** Hyun-Ok Jung, Seung-Woo Han.

**Methodology:** Seung-Woo Han.

**Supervision:** Seung-Woo Han.

**Visualization:** Seung-Woo Han.

**Writing – original draft:** Hyun-Ok Jung, Seung-Woo Han.

**Writing – review & editing:** Hyun-Ok Jung, Seung-Woo Han.

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
